# Placebos in primary care? a nominal group study explicating UK GP and patient views of six theoretically plausible models of placebo practice

Mohana Ratnapalan [iD],[1] Beverly Coghlan,[2] Mengxin Tan,[3,4] Hazel Everitt,[5] Adam W A Geraghty [iD],[1] Paul Little,[1] George Lewith,[1] Felicity L Bishop[1,6]

Lewith died on 17 March 2017

This paper presents independent research funded by the National Institute of Health Research (NIHR).

For numbered affiliations see end of article.

**Correspondence to**
Dr Felicity L Bishop;
F.L.Bishop@soton.ac.uk

## ABSTRACT

**Objectives** To better understand which theoretically plausible placebogenic techniques might be acceptable in UK primary care.

**Design** A qualitative study using nominal group technique and thematic analysis. Participants took part in audio-recorded face-to-face nominal groups in which the researcher presented six scenarios describing the application in primary care of theoretically plausible placebogenic techniques: (1) Withholding side effects information, (2) Monitoring, (3) General practitioner (GP) endorsement, (4) Idealised consultation, (5) Deceptive placebo pills and (6) Open-label placebo pills. Participants voted on whether they thought each scenario was acceptable in practice and discussed their reasoning. Votes were tallied and discussions transcribed verbatim.

**Setting** Primary care in England.

**Participants** 21 GPs in four nominal groups and 20 'expert patients' in five nominal groups.

**Results** Participants found it hard to decide which practices were acceptable and spoke about needing to weigh potential symptomatic benefits against the potential harms of lost trust eroding the therapeutic relationship. Primary care patients and doctors felt it was acceptable to harness placebo effects in practice by patient self-monitoring (scenario 2), by the GP expressing a strongly positive belief in a therapy (scenario 3) and by conducting patient-centred, empathic consultations (scenario 4). Deceptive placebogenic practices (scenarios 1 and 5) were unacceptable to most groups. Patient and GP groups expressed a diverse range of opinions about open-label placebo pills.

**Conclusions** Attempts to harness placebo effects in UK primary care are more likely to be accepted and implemented if they focus on enhancing positive patient-centred communication and empathic relationships. Using placebos deceptively is likely to be unacceptable to both GPs and patients. Open-label placebos also do not have clear support; they might be acceptable to some doctors and patients in very limited circumstances—but further evidence, clear information and guidance would be needed.

### Strengths and limitations of this study

► Nominal group technique and thematic analysis were used to identify key opinions from both general practitioners and patients on theoretically plausible placebogenic techniques. Participants were recruited through research networks and patient charities and sampled to achieve a broad range of views.

► Scenarios discussed were carefully constructed to reflect potential placebogenic practice based on clinical and experimental research evidence.

► Some nominal groups were small due to the availability of participants.

► It was not always possible to achieve a clear majority opinion on the scenarios.

## BACKGROUND

Placebos have an uncertain role in everyday medical practice. They have a long history[1–3] and evidence that suggests therapeutic benefit.[4–10] However, there is no broad consensus on how to define placebos nor the ethics of use in clinical practice.[11 12] Definitions vary between placebos as a substance, a process (eg, practitioner empathy) or both.[11] This paper defines placebos as substances or processes other than the active ingredients of treatment, which can have substantial effect on symptoms. We define placebo effects as beneficial symptomatic changes triggered by the meaning a person experiences in a healthcare setting.[13 14]

In the UK, there are over 300 million primary care consultations annually[15] with rising demand in the last decade.[16] Within this context, it becomes important to optimise doctor-patient encounters for maximum health benefit. Placebogenic practices, that is, techniques that can trigger placebo responses in clinical settings, could support cost-effective healthcare, which minimises patient harm from drug side effects and/or

enhances the effects of prescribed evidence-based therapies. A recent meta-analysis describes frequent use of placebos in primary care with particularly high use of non-specific therapies (eg, physician as placebo to exert positive psychological effect).[17] A meta-ethnographic review of patient and doctor views on placebo practice found acceptable use to patients include therapeutic benefit and giving hope; with healthcare professionals also citing therapeutic benefit and placebos as clinical management tools.[11] However, few studies directly compared doctors' and patients' views. A meta-analysis on open-label placebos, where patients are honestly informed they are being given placebos found positive clinical effects.[8] However, few qualitative studies have explored patients' or doctors' perspectives on open-label placebos.[18] We used nominal group technique,[19–21] a qualitative consensus building technique, to explore and compare how patients and doctors conceive a range of placebogenic practices and why certain practices are more acceptable.

## METHODS

Our research team consists of health psychologists (FB, AG), general practitioners (GPs) (PL, GL, HE), a psychotherapist (BC), a psychology student (MT) and a GP Academic Clinical Fellow (MR). This range of backgrounds enriched our qualitative analysis, enabling us to bring diverse perspectives to the data and ensuring we explored multiple potential themes, remained open to new ways of conceptualising the data, worked to evidence our interpretations in the raw data and avoided an idiosyncratic interpretation. Ultimately our approach to data collection and analysis was driven by our pragmatic aim to examine which placebogenic practices might be more or less acceptable to patients and GPs and why.

### Patient and public involvement

No patient involved. We did not specifically involve patients or the public in the design, conduct or reporting of this study. However, this study aimed to capture patient views.

### Participant recruitment

We recruited English-speaking GPs and adult 'expert patients', that is, those with recent experience of using health services and involvement in patient organisations or medical research. We advertised to GPs through the south-west primary care research network and to patients through UK-wide patient associations and health charities (eg, Pain UK, groups for people with Chronic Fatigue Syndrome and similar conditions). Individuals expressing interest were mailed a participant information sheet and completed a consent form before participation. We deliberately sought to include GPs and patients of a range of ages and genders and patients with a range of health conditions.

Participants who agreed to participate did so on prespecified days. The number of people willing to participate determined group size and composition.

### Nominal groups

We structured nominal group meetings as per methodological guidance[19 21] (table 1). BC, an accredited psychotherapist experienced in facilitation, led the group meetings held in suitable venues (eg, meeting rooms in GP practices) between April 2013 and August 2013.

Nominal groups were presented six scenarios for voting and discussion. These scenarios were written by the research team using a taxonomy of five domains of placebogenic techniques[22–26] derived from experimental and clinical studies[27] to create six theoretically plausible placebogenic scenarios for primary care (table 2). Techniques from the five domains[22] were

| Table 1 | Nominal group meeting structure |
|---|---|
| **Phase** | **Activity** |
| Informed consent | Facilitator (BC) talks through participant information sheets and consent forms and answers any questions. Participants sign consent forms. |
| 1: Introduction | Facilitator introduces the topic, explains our interest in it and asks participants to introduce themselves. |
| 2: Silent reflection | Participants read the scenario and write comments. |
| 3: Round robin | Facilitator elicits one comment from each participant and writes this on a flip chart. Discussion not allowed. Continues until comments exhausted. |
| 4: Discussion | Facilitator guides discussion of comments on each scenario in turn, using open-ended questions and ensuring all participants had the opportunity to contribute their perspective. |
| 5: Voting | Participants vote whether the scenario is acceptable or not. (undecided was also permitted) |
| 6: Repeat | Processes 2 to 5 are repeated in turn for each scenario. |
| 7: Break | Facilitator counts votes. |
| 8: Discussion | Results of votes presented and discussed. Each scenario without clear majority is discussed in turn. |
| 9: Voting | Second round of voting if no clear majority in first round of voting. |
| 10: Conclusion | Results of vote. Facilitator explains future plans and thanks participants. |

**Table 2** Scenarios for patient groups

| Scenario | Aspect that might enhance placebo responding |
|---|---|
| **Scenario 1: 'Withholding side effects'** You visit your GP because you have noticed new or worsened symptoms. Your GP examines you, asks you about your symptoms, gives you a diagnosis and decides to prescribe medication. Your GP knows that if she provides you with positive information about the medication you are more likely to notice a benefit. So to make you feel hopeful about your treatment she tells you, truthfully, that research has shown that the majority of patients taking this medication notice a big improvement in their symptoms, and that you too, should notice a big improvement. The medication might have side effects, but your GP does not tell you about these. This is because she knows that if she *does* tell you about the possible side effects then you will be more likely to suffer from them. | Giving a positive message may enhance patients' response expectancy; withholding information about medication side-effects may reduce the chances of the patient developing them via nocebo mechanisms. |
| **Scenario 2: 'Monitoring'** You visit your GP because you have noticed new or worsened symptoms. Your GP advises you to continue with your usual treatment but requests that you attend the surgery more frequently for ongoing review and monitoring of your condition. She also asks you to monitor your symptoms on a daily basis and report back to her at your next visit. She provides you with a special symptom-monitoring diary to help you to do this. | The use of regular monitoring and review may increase awareness of symptom changes and potentially motivate behavioural changes. |
| **Scenario 3: 'GP endorsement'** You visit your GP because you have noticed new or worsened symptoms. Your GP examines you, asks you about your symptoms, gives you a diagnosis and offers to prescribe a particular medication. You have heard of this medication and are not sure how effective it will be and ask if there are any other treatments you could try instead. Your GP says that there are but that he strongly believes (based on his experience with other patients and from published research) that the medication he wants to prescribe provides absolutely the best chance of reducing your symptoms in the shortest time. | Conveying the clinicians' strong personal beliefs about a particular medication may enhance patients' response expectancy. |
| **Scenario 4: 'Idealised consultation'** You visit your GP because you have noticed new or worsened symptoms. Your appointment is with the same GP you always see. He greets you warmly and seems pleased to see you. He turns away from his computer screen and gives you his full attention. He is very interested and concerned about what you tell him. He asks you many detailed questions about how the symptoms started and how they are now affecting your daily life. He thoroughly examines you. He genuinely seems to be interested in you as a person and not as just a collection of symptoms. He allows you time to ask questions and even though he does not know all of the answers he provides as much information as he can and says he will try to find out more and will get back to you later in the day by telephone. He tells you that he would prefer it if you continue to make your appointments with him in future. | Enhanced attention, more time, warm and empathic and collaborative style may enhance perception of empathy, validation and response expectancy. |
| **Scenario 5: 'Deceptive placebo pills'** You attend your GP surgery because you have noticed new or worsened symptoms. Your GP examines you, asks you about your symptoms, gives you a diagnosis and recommends a prescription for medication. She tells you that research has shown that by taking this medication three times a day for at least a week, your symptoms will get better. What she **does not** tell you is that the medication she will be prescribing is actually a 'placebo' pill that contains **no real medicine**. | Prescribing a placebo medication deceptively may enhance response expectancy and engender a conditioned response to pill taking. |

Continued

| Table 2 | Continued |

| Scenario | Aspect that might enhance placebo responding |
|---|---|
| **Scenario 6: 'Open-label placebo pills'**<br>You attend your GP surgery because you have noticed new or worsened symptoms. Your GP examines you, asks you about your symptoms, gives you a diagnosis and recommends a prescription for medication. She tells you that research has shown that by taking this medication three times a day for at least a week, your symptoms will get better. What she **does** tell you is that the medication she will be prescribing is actually a 'placebo' pill that contains **no real medicine**. | Prescribing a placebo medication openly may enhance response expectancy and engender a conditioned response to pill taking. |

GP, general practitioner.

used to create the scenarios: (1) The patient's beliefs and characteristics informed 'Withholding side effects', (2) The healthcare setting informed 'Monitoring', (3) The practitioner's beliefs and characteristics informed 'GP endorsement', (4) The patient-practitioner interaction informed 'Idealised consultation' and (5) Treatment characteristics informed 'Deceptive'/'Open-label placebo pills'. GP groups read scenarios written from the GP perspective.

### Data analysis

Each meeting was audio-recorded, transcribed verbatim and anonymised.

Our analytical approach rested on principles described by McMillian *et al* and encompassed attending to both participants' votes and qualitative discussions.[20] Votes were counted and each group was classified according to whether the majority of participants deemed each scenario acceptable, unacceptable or 'no clear majority'. To analyse the discussions we used thematic analysis[28] with constant comparison between groups and scenarios. After repeated reading of transcripts, initial low-level inductive codes were developed independently for the GP and patient transcripts by MR and MT, respectively, using NVivo 12 to facilitate coding and maintain an audit trail. Low-level codes were reviewed by FB, MR and MT who iteratively developed higher level codes by merging similar low-level codes and combining them into a hierarchical structure. MR led the search for themes by comparing and contrasting codes across scenarios and across GP and patient groups, reviewing potential themes for fit with the raw data. MR, HE and FB discussed which themes best captured GPs' and patients' reasoning around placebogenic practice and agreed on the final 16 themes (see online supplementary appendix 1). MR then integrated the qualitative themes with the vote frequency data using an iterative process comparing votes and themes (a) within groups across individual scenarios and (b) within scenarios across groups. This analysis was developed and agreed by all authors and is presented below. We used the Standards for Reporting Qualitative Research checklist when writing our report.[29]

### RESULTS

### Participant characteristics

Twenty-one GPs and twenty patients (table 3) participated in nine nominal groups (four GP and five patient groups); with two to eight participants and lasting 75 to 100 min per group. Most GPs (n=17, 81%) were working full-time. Two patients completed sixth form or college, four university undergraduate and nine post-graduate (five did not disclose). Fifteen patients disclosed their general health status as follows: very good, n=7 (35%); good n=1 (5%); fair n=6 (30%) and bad n=1 (5%). Patients' self-reported health conditions included: chronic pain, irritable bowel syndrome, cancers and diabetes.

### Qualitative analysis

#### Overview

Participants found it hard to decide whether each placebogenic practice was acceptable. Patients and GPs spoke about the tension between balancing positive effects of placebogenic practice against harmful erosion of the therapeutic relationship from loss of trust.

> "But I think you have got to be so careful… because the breach of trust and that feeling of breach of trust, can have worse effects I think than the positive effect… so it is a balancing act."

| Table 3 | Demographics | | |
|---|---|---|---|
| | | **GP** | **Patient** |
| Total n | | 21 | 20 |
| Number of males (%) | | 12 (57%) | 7 (35%) |
| Number of females (%) | | 9 (43%) | 13 (65%) |
| Mean age (SD)* | | 42 (9.2) | 56.3 (12.7) |
| Mean years GP (SD)† | | 15 (10.1) | – |
| Range of group size (mean) | | 3–8 (5) | 2–7 (4) |

Undisclosed demographic data comes from different nominal groups and is not isolated missing data for any single group.
*5 not disclosed.
†3 not disclosed.
GP, general practitioner.

**Table 4** Tabulated group level voting on acceptability of six scenarios of placebogenic practice

| Scenario | Acceptable | No clear majority | Unacceptable |
|---|---|---|---|
| 1. 'Withholding side-effects' | Δ Δ | Δ Δ ○ | Δ ○ ○ ○ |
| 2. 'Monitoring' | Δ Δ Δ Δ ○  ○  ○  ○ | Δ | |
| 3. 'GP endorsement' | Δ Δ Δ ○  ○  ○  ○ | Δ | |
| 4. 'Idealised consultation' | Δ Δ Δ Δ ○  ○  ○  ○ | | |
| 5.'Deceptive placebo pills' | | Δ | Δ Δ Δ Δ ○  ○  ○  ○ |
| 6.'Open-label placebo pills' | ○  ○ | Δ Δ Δ ○ | Δ Δ ○ |

○=GP groups (n=4), Δ=Patient groups (n=5).
GP, general practitioner.

(Patient Group 1)

"… the nice thing about GPs is having the ongoing patient relationship. So we' re also trying to build a relationship and that's, obviously, part of a placebo effect. But if you tell patients it's going to work brilliantly and it doesn't then that slightly damages their trust, vs if you tell them that they might get a side effect but it will settle down… But again, it's either damaging or enhancing in the GP relationship, as well."

(GP Group 3)

Despite these tensions, there were some consistent patterns in the voting (table 4). 'Monitoring' and 'GP endorsement' were acceptable to all GP groups while the 'Idealised consultation' was acceptable to all GP and patient groups. The arguments that participants offered in the discussions to justify their votes are explored below.

GPs and patients felt that 'Monitoring' empowered patients by providing patient-centred care. GPs argued that involving the patient and using time as diagnostic tool could help them consult more effectively, but expressed concern that overemphasising symptoms could lead to psychological harms (eg, generating anxiety). The acceptability of this scenario was felt to depend on the medical condition, the patient's characteristics (eg, age), the work required of the patient for self-monitoring, the disease process and the symptom severity.

"…it would also provide me with more control… over my condition… by being aware of change."

(Patient Group 5)

"Potentially, I'll say, I would do this—if, in the 10 min I've got available, I really haven't got a true reflection of you know, what the symptom pattern is and what the effect on the patient really is, then it's just a way of extending the consultation over a period of time and to actually gather that information."

(GP Group 2)

GP groups discussed how the GPs' experience and the evidence-base would influence the acceptability of 'GP endorsement'. GPs felt there needed to be a published evidence-based benefit or personal experience of likely therapeutic benefit to endorse a therapy. Patients felt that 'GP endorsement' might be more acceptable in the context of more egalitarian doctor-patient relationships.

"Again I think it depends on the relationship I as the patient had with that GP, whether it was a relationship I felt was equal, or not. If it was then I would be more inclined to go along with that advice. If I felt it was more of a paternalistic relationship than I would be questioning why, why does he think this is the best one. I'd want more information about that drug, and also to discuss whatever it is that I've heard about this medication and why I've heard it's not necessarily the best thing. And also to be sure that it's not being prescribed because it's the latest drug that pharmaceuticals are pushing and this is a really good one and it will do all singing, all dancing."

(Patient Group 2)

The continuity of care within the 'Idealised consultation' was particularly well-received. GPs felt continuity of care enhanced their job satisfaction and improved their understanding the psychosocial context of their patients by permitting long-term relationships to develop with patients and families. GPs felt continuity provided a directed trajectory of care that disjointed multi-practitioner led care might not provide. Patients agreed and valued the idea of seeing a practitioner who knew their story.

Despite universal acceptance of the 'Idealised consultation', GPs and patients also expressed concerns about this. GPs were concerned that knowing their patients too well could lead to harm from cognitive bias and encourage patients to become overly reliant on one doctor and subsequently come to harm from delaying presenting if that

doctor was unavailable. GPs and patients both expressed concerns that this scenario was unrealistic given workloads and/or would increase GP workload, which may in turn negatively affect care, and that this type of practice could blur doctor-patient boundaries.

"And sometimes you know your patient so well that you just don't see that they're losing weight or they're becoming hypothyroid or something."

(GP Group 4)

" (F1): Well, I say a GP from heaven!… I would have full trust and confidence in the GP, if ever they had that sort of response to you and a welcoming aspect to it, and naturally being eye-contact, focussing with you, and receptive both ways. And interested in you and there is communication, as a key factor. And to be able to leave that surgery knowing that you have got some form of support out there, in such an isolating situation whenever you are in chronic pain.

(F3): I mean I think even the admission that he doesn't know all the answers is reassuring, because GPs are what GPs are, they are not specialists, they have to know something about a lot of things, but not necessarily deep down into one specialisation, but they know where to go…"

(Patient Group 1)

### Deception in placebogenic practice

'Withholding side effects' and 'Deceptive placebo pills' both involve deception. Most groups found 'Withholding side effects' unacceptable or impossible to reach a majority judgement, while all but one group found 'Deceptive placebo pills' unacceptable.

GPs and patients were worried about the risks of physical and psychological harm and damage to the GP-patient relationship from withholding information about side effects. For example, one GP group was concerned about patient harm from an accident if they were unaware of potential impaired function. One patient group discussed how an unexpected side effect might cause anxiety that this was a new health problem. Patients felt that withholding information disempowers them and being inconsistent with patient-centred care where ultimate autonomy rests with the patient to make informed decisions. GP groups also discussed medico-legal and policy issues. They worried about medico-legal implications of non-disclosure and discussed how government targets may alter their discussions about medication. However, patients were more accepting of 'Withholding side effects' than the GP groups. The patient groups who found this to be acceptable practice spoke about GPs knowing their patients and using their judgement on when it might be permissible to not mention side effects based on having an effective partnership built on trust with their GP.

"Yes, I think my views change with time, too, and the outside world we're working in. I'm far more likely to give somebody an ACE inhibitor now than I was 5 years ago, simply because its part of the QOF targets. And that has a big effect on the way I will sell an ACE, give an ACE and encourage the patient to use it. I probably use quite a lot of the 'Doctor knows best' concept in the consultation to push that particular drug. Because there is a monetary target involved with it."

(GP Group 3)

"To me, it depends on the frequency and the severity of the side effects. Because if they're rare and minor I would be completely comfortable with it, if they're serious or very frequent I'd be uncomfortable with it because you risk loss of trust… And again, there's the risks, especially if… it impaired their function and they had an accident or, you know, there's risks to not telling people about potential side effects."

(GP Group 1)

"Yes. If you're expecting a patient to take a drug then they have to understand potentially what the problems or issues could be. I mean, you know, even if they are fairly minor I think most people understand that it's only a 1% chance of things happening, but at least it's their decision to take that drug or to take that treatment, and they can't take that decision if they're being pushed, if you like, being pushed or being persuaded to do it if they don't get the full information and it's not the doctor's decision, it's the patient's decision."

(Patient Group 4)

In contrast, 'Deceptive placebo pills' involved more active deception and was felt to be dishonest. GPs were concerned that this was ethically unacceptable practice and this drove their decision-making irrespective of potential benefit from placebo pills. GPs felt that it was imperative that patients were able to make fully informed decisions about therapy in this scenario. There was fear of repercussions from the General Medical Council, as the use of placebo pills is not, currently, incorporated into professional codes of practice nor accepted within the wider cultural context of medical practice. One GP group felt there might be a role for deceptive placebo pills in (unspecified) mental illness. However, the same group expressed a tension between personal ethics and accepted codes or standards of practice.

Similarly, patients expressed discomfort with receiving unknown substances and judged the deceit involved as ethically unacceptable. Patients spoke of risk of psychological harm from feeling that their symptoms were not 'real' and were 'all in the mind', with some seeing placebo pills as not real therapy for real symptoms. They also worried that placebo pills would be a way for GPs to avoid properly investigating their problems. Patients spoke of subverting the placebogenic effect by seeking information about the pills outside the consultation (eg, online). Both patients

and GPs expressed concerns about deceptive prescribing eroding the doctor-patient relationship.

"(M1): Do you believe in a placebo?

(F1): …I do believe in it but that's not actually what's being asked… I believe that the patient should know what it is they're signing up to. So I'm really happy in a proper clinical trial where you're told you could go into the placebo arm or you could go into the arm where you will get the drug. That's absolutely fully acceptable and you then don't know whether you've got the placebo. That's great, but this is wrong, this is underhand.

(F2): I think it's the GP judging you, thinking he or she knows you really well. I mean how well do they know you from a 5 to 10 min consultation. You know, have they asked you about other things going on in your life? Other issues and things, or are they just focussing on that one aspect.

(F1): Yeah, give them a dummy pill and then they'll go away and be quiet. As opposed to actually, you know, getting down to what the issue actually is."

(Patient Group 4)

"And how do you feel about that? How do you defend against that? And what is the patient going to think of you next time they see you, if they realise it was a fake pill, so to speak? And how do they have confidence in you from thereon in?"

(GP Group 2)

"I could live with it ethically, I think the problem is the GMC code of practice, isn't it?"

(GP Group 1)

### Open-label placebo pills

In contrast to 'Deceptive placebo pills', 'Open-label placebo pills' removed the element of deception with placebo pills. Despite many of the objections to 'Deceptive placebo pills' focussing on the deception per se, removing the deceptive element did not lead to a complete shift towards groups seeing placebos as acceptable. Although GPs groups were more accepting of this scenario, patients felt that the acceptability of open-label placebo depended on the medical condition and their trust in the GP. Patients were not happy to pay a prescription charge for what they saw as an inert pill and if they saw placebo as inappropriate or ineffective they argued that this would weaken the therapeutic effect.

"That it was a placebo and that it was found to work in other people, I'd think great, I'll give it a go, yeah. I'd be quite happy about that, it's the not being told that I have the problem with."

(Patient Group 2)

Some GPs felt that prescribing alone was not enough and additional positive talk and a cultural shift would be required. Others worried that patients would stop seeing them if they prescribed placebo pills.

"(M1) …I've never personally done that but I know when I was doing paediatrics there was a child with quite profound functional symptoms… and they knew they were having oral saline and they got better, they improved. So I would be more comfortable with that but I've never had a clinical context where I've had the courage to do it but if it…

(F2) Me neither.

(M1) But if it was more of a sort of cultural thing I would be very glad we're doing that."

(GP Group 4)

"I just think it's mad. If I did that with my patients they'd never come and see me again and say, "I'll get a doctor that gives me actual medicine"."

(GP Group 1)

## DISCUSSION

Our study captures both GP and patient views and offers new insights on the real-world application of placebos. We found that placebogenic practice with deception is very clearly not acceptable and for open-label placebo pills, there was no clear judgement of acceptability from any of the patient groups. This extends on previous studies, which suggest that GPs found deceptive placebogenic practice unacceptable[30 31] and some patients feel it is important for any placebogenic practice to respect patient autonomy.[32] By focussing on theoretically plausible placebogenic scenarios, we provide new insights to placebogenic practices with potential for implementation to enhance patient outcomes in clinical practice and clarified the psychological and sociocultural barriers that would need to be overcome.

We found the acceptability of placebogenic practice is difficult to determine and even 'acceptable' scenarios elicited talk of caveats, as did a recent meta-ethnography.[11] Caveats to acceptability identified in our study include, but are not limited to, considerations of: medical condition, individual patient, individual doctor, regulatory norms, government prescribing targets, General Medical Council (GMC) guidance and what is viewed as acceptable practice to other medical colleagues (ie, social norms). This suggests that any generic guidelines proclaiming a type of practice as either 'acceptable' or 'unacceptable' may not capture the views of GPs and patients as key stakeholders and may be problematic. It may be more useful to develop guidance that highlights important considerations and contexts for placebogenic practice.

Our study is limited by non-blinded voting. The group method means discussions must be interpreted in their social and cultural context, and not as individuals' personal beliefs. It is informative, although not surprising, that the GP groups discussed clinical practice norms while the patient groups (comprising expert patients

who were typically accustomed to acting as patient advocates) drew heavily on the notion of patient autonomy. Indeed, the composition of our patient groups must be considered when interpreting our findings. By deliberately seeking 'expert patient' participants we have gained insight into how particularly engaged and politically aware patients with high health literacy discuss placebos in general practice. Had we sought a more diverse sample of patients including for example those with lower health literacy and less engagement with health services then different issues may have emerged as important in the patient group discussions. Despite attempts to purposively sample a diverse group, some nominal groups were small with two or three participants. Our findings may also be limited by the sequence in which cases were presented to groups. Participant views on open-label placebo may have been influenced by preceding discussions of placebo pills prescribed deceptively. However, it was felt important to present scenarios in a way that would encourage discussion and offer participants a 'way in' to this complex topic, hence we chose to present the more familiar examples of deceptive placebo prescribing before moving on to explore open-label placebo. Findings indicate patterns of views held by our participants, all of whom volunteered their time and so might be more interested in and/or hold stronger views about placebo effects than non-participants.

## CONCLUSIONS AND FUTURE WORK

Our study helps inform future work on placebogenic practice and provides clinicians with improved understanding of what peers and patients would find acceptable, while acknowledging this is a complex area with diverse opinions. Our study suggests that open-label placebo pills are not fully acceptable and future translational work could consider prescription costs among other issues. Additional research evaluating acceptable placebogenic techniques in clinical settings is needed to help inform clinicians about the effectiveness of these techniques in clinical practice.

**Author affiliations**
[1]Primary Care and Population Sciences, University of Southampton, Southampton, UK
[2]ACT Works Ltd, Portsmouth, UK
[3]Centre of Global Mental Health, London School of Hygiene and Tropical Medicine, London, UK
[4]Institute of Psychiatry, Psychology and Neuroscience, London, UK
[5]Primary Care and Population Sciences, Faculty of Medicine, University of Southampton, Southampton, UK
[6]Centre for Clinical and Community Applications of Health Psychology, University of Southampton, Southampton, UK

**Contributors** FB, GL, PL, BC, HE and AG conceived and designed the study. BC conducted the nominal group meetings. FB, BC, HE, MT and MR were involved in the data analysis and interpretation. MR led the integrated analysis and drafting of the manuscript. All authors contributed to the drafting of the manuscript and take responsibility of the integrity of the data and the analysis.

**Funding** The project 'Creating a Taxonomy to Harness the Placebo effect in UK primary care' was funded by the National Institute of Health Research (NIHR) School for Primary Care Research (SPCR) (project number 161). Additional funding for BC was provided by Solent NHS Trust. MR also received funding for part of her research time from the SPCR.

**Disclaimer** The views expressed are those of the author(s) and not necessarily those of the NHS, the NIHR, HEE or the Department of Health. The funders had no role in design and conduct of the study; collection, management, analysis and interpretation of the data; preparation, review or approval of the manuscript and decision to submit the manuscript for publication.

**Competing interests** None declared.

**Patient consent for publication** Not required.

**Ethics approval** Ethical approval for this study was obtained from the Faculty of Medicine Ethics Committee, University of Southampton (reference number: 4741). Participants gave their informed consent before taking part.

**Provenance and peer review** Not commissioned; externally peer reviewed.

**Data availability statement** Data are available upon reasonable request. Qualitative coding data available in Appendix 1. Further quotations may be available, subject to ethical approval, by contacting the authors.

**ORCID iDs**
Mohana Ratnapalan http://orcid.org/0000-0002-6505-6107
Adam W A Geraghty http://orcid.org/0000-0001-7984-8351

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
