## [Reviewer comments · BMJ Open]

ARTICLE DETAILS

TITLE (PROVISIONAL)	Placebos in Primary Care? A Nominal Group Study Explicating UK GP and Patient Views of Six Theoretically Plausible Models of Placebo Practice
AUTHORS	Ratnapalan, Mohana; Coghlan, Beverly; Tan, Mengxin; Everitt, Hazel; Geraghty, Adam; Little, Paul; Lewith, George; Bishop, Felicity

VERSION 1 - REVIEW

REVIEWER	Steven Savvas, Research Fellow National Ageing Research Institute, Australia
REVIEW RETURNED	22-Jul-2019

GENERAL COMMENTS	Dear Authors, Your study investigating the acceptable of theoretically plausible placebogenic techniques is interesting, relevant and topical. I have only a few minor comments. Pg4 line 6 and Pg12 line 45. I think a major limitation of the study is the use of 'expert' patients. In patient recruitment (pg 4), more detail on the 'expert patients' would benefit the reader (eg. a few have been participants in research studies? whilst most work for patient advocacy?) And it is likely that these 'expert' patients have very high health literacy and as they are advocates I am unsure how much their views reflect the general population. I don't think a comment in parenthesis that is phrased as an explanation for a finding (pg 12, line 46) gives sufficient weight to this limitation. Pg4 line 59 and pg5 line29. For technique '2: Monitoring' and table 2. I'm unfamiliar with the research that some of your 'monitoring' approaches are an effective placebogenic technique (esp self-monitoring, symptom diaries). The reference given (Di Blasi et al 2001) does not talk specifically about it. Do you have any references you can add here? Pg 7 Table 4. i agree with your interpretation of scenario 2,3,4 and 5. My impression of scenario 1 is that patients are more accepting than GPs (as only 1 patient group found it unacceptable whilst no GP group found it acceptable), and that for scenario 6 GPs were more accepting than patients (as only 1 GP group found it unacceptable whilst no patient groups found it acceptable). However the results don't really reflect this interpretation. On pg9 line 49-59, there is no reporting of how 2 of the patient groups
--

	found scenario 1 acceptable (which presumably at least half of the patients found it acceptable [unless the unequal group sizes confounded the results?]). Likewise, pg11 line36 doesn't report that GPs were more accepting than patients. And it is unclear what position the quote on pg11 line 43 supports, 'Then i'd want my 7pounds...' Is this a pro or against quote? Appendix 1: some example quotes are too fragmented and may need some expansion. eg pg18 line 19. 'I suppose i would use it...' Use what? A symptom diary? eg pg19 line 31. 'if they're serious or very frequent...' i assume you mean 'if they're serious or very frequent [symptoms]...'?
--	--

REVIEWER	Patrick Finan Johns Hopkins University School of Medicine I have an ongoing placebo-related research project that I am exploring for commercial potential, but does not at present have any commercial interests.
REVIEW RETURNED	03-Oct-2019

GENERAL COMMENTS	This manuscript describes a qualitative study that assessed patient and general practitioner views on placebos (both deceptive and open-label) and other practices involving varying degrees of provider-patient communication. The authors report that there were consensus views in both groups (patients and GPs) that deceptive placebos are unacceptable in medical practice. The views were mixed in both groups regarding open-label placebos, with some finding the practice acceptable under certain conditions of provider-patient communication, and others finding it unacceptable altogether. The manuscript is interesting and appears to follow sound qualitative methods (this is not my area of expertise, and so I can only comment generally here). It addresses an important and very current topic that is increasingly working its way into discussions of medical ethics, as more open-label placebo trials are published. The writing is clear and the manuscript could potentially make a useful contribution to the literature. However, the following issues should be addressed:  1) Please give the frequencies associated with each of the self-reported health conditions. 2) Please give a breakdown of patient race. 3) Was there any additional patient info available related to medications currently being used? If so, this would be helpful to contextualize their responses. 4) It would be important to know whether the patients sampled get their care primarily through GPs or perhaps through tertiary care providers. Is that information available? It is relevant to placing patient responses into the context of GP responses. 5) The authors state that the sample size was determined by the number of interested participants. This seems very post hoc. Was
---

	there a plan at the outset for a desired number of participants? It is difficult to ascertain from the manuscript whether the final sample size is sufficient for the purpose of the study. A stronger, preferably a priori, justification of sample size is needed. 6) It is not clear from the manuscript whether the scenarios were discussed with participants sequentially or randomly. If sequentially, then it should be described as a potential limitation of the results. Participants' views of open-label placebo could have been negatively biased by their discussion of deceptive placebos if the open label discussion immediately followed the deceptive placebo discussion. A random presentation of the various scenarios would have been preferable.
--	--

VERSION 1 – AUTHOR RESPONSE

Reviewer 1:

1. Pg4 line 6 and Pg12 line 45. I think a major limitation of the study is the use of 'expert' patients. In patient recruitment (pg 4), more detail on the 'expert patients' would benefit the reader (eg. a few have been participants in research studies? whilst most work for patient advocacy?) And it is likely that these 'expert' patients have very high health literacy and as they are advocates I am unsure how much their views reflect the general population. I don't think a comment in parenthesis that is phrased as an explanation for a finding (pg 12, line 46) gives sufficient weight to this limitation.

Thank you for highlighting this point, we have added a more detailed consideration of this limitation to page 12: “Indeed, the composition of our patient groups must be considered when interpreting our findings. By deliberately seeking “expert patient” participants we have gained insight into how particularly engaged and politically aware patients with high health literacy discuss placebos in general practice. Had we sought a more diverse sample of patients including for example those with lower health literacy and less engagement with health services then different issues may have emerged as important in the patient group discussions.”

2. Pg4 line 59 and pg5 line29. For technique '2: Monitoring' and table 2. I'm unfamiliar with the research that some of your 'monitoring' approaches are an effective placebo technique (esp self-monitoring, symptom diaries). The reference given (Di Blasi et al 2001) does not talk specifically about it. Do you have any references you can add here?

Thank you for highlighting this. We have added citations to clarify this point.

3. Pg 7 Table 4. i agree with your interpretation of scenario 2,3,4 and 5. My impression of scenario 1 is that patients are more accepting than GPs (as only 1 patient group found it unacceptable whilst no GP group found it acceptable), and that for scenario 6 GPs were more accepting than patients (as only 1 GP group found it unacceptable whilst no patient groups found it acceptable). However the results don't really reflect this interpretation. On pg9 line 49-59, there is no reporting of how 2 of the patient groups found scenario 1 acceptable (which presumably at least half of the patients found it acceptable [unless the unequal group sizes confounded the results?]). Likewise, pg11 line36 doesn't report that GPs were more accepting than patients. And it is unclear what position the quote on pg11 line 43 supports, 'Then i'd want my 7pounds...' Is this a pro or against quote?

Thank you for this important point on our interpretation of the scenarios.

We have added the following text to better capture the views on scenario 1 (pg 9):

“However, patients were more accepting of ‘Withholding side effects’ than the GP groups. The patient groups who found this to be acceptable practice spoke about GPs knowing their patients and using their judgement on when it might be permissible to not mention side effects based on having an effective partnership built on trust with their GP.”

We have also expanded scenario 6(p 11) with the following:

“Although GPs groups were more accepting of this scenario, patients felt that the acceptability of open-label placebo depended on the medical condition and their trust in the GP. ”

We have deleted the quote as it is described in the text and hope this improves the clarity of the section.

4. Appendix 1: some example quotes are too fragmented and may need some expansion. eg pg18 line 19. 'I suppose i would use it...' Use what? A symptom diary? eg pg19 line 31. 'if they're serious or very frequent...' i assume you mean 'if they're serious or very frequent [symptoms]...'?

Thank you for reviewing our appendix and selection of quotes. We have expanded on the highlighted quotes as described below. Our authorship team has also reviewed the appendix section and updated this to improve clarity with our example quotes.

pg18 line 19. 'I suppose i would use it...' Use what? A symptom diary?

- We have added a parenthetical explanation to signpost reader to which scenario this quote refer
eg pg19 line 31. 'if they're serious or very frequent...'

We have expanded this quote as follows: “To me, it depends upon the frequency and the severity of the side effects. Because if they’re rare and minor I would be completely comfortable with it, if they’re serious or very frequent I’d be uncomfortable with it because you risk loss of trust, I think, from your patient if you don’t tell them.”

Reviewer 2:

1. Please give the frequencies associated with each of the self-reported health conditions.
2. Please give a breakdown of patient race.
3. Was there any additional patient info available related to medications currently being used? If so, this would be helpful to contextualize their responses.

We thank reviewer 2 for their comments and would like to address the first three points made together. Each of the self-reported conditions are from 1 to 2 participants. We appreciate the value added from further contextual information such as patient race and other medications prescribed. We sought to seek views from a range of different individuals who have experience of consulting with their GP. However, with a small sample of patients with particular combinations of illnesses we prefer not to risk breaching anonymity and therefore retain reporting at the group level, as per good practice in small sample qualitative work. Thus, whilst we appreciate the feedback from reviewer 2 regarding further detail on the participants, given the low frequency of each health conditions and the desire to preserve anonymity we have respectfully elected not to add this information to our paper.

4. It would be important to know whether the patients sampled get their care primarily through GPs or perhaps through tertiary care providers. Is that information available? It is relevant to placing patient responses into the context of GP responses.

Thank you for this comment. In the UK, 90% of patient contact with the National Health Service (NHS) is via primary care. We also purposefully recruited patients who had received care via their GPs and the group discussions typically focused on the patients' experiences of GP care. Patients only occasionally drew on their experiences of secondary care (e.g. pain clinics). We have thus, respectfully, not elaborated in our manuscript on this point.

5. The authors state that the sample size was determined by the number of interested participants. This seems very post hoc. Was there a plan at the outset for a desired number of participants? It is difficult to ascertain from the manuscript whether the final sample size is sufficient for the purpose of the study. A stronger, preferably a priori, justification of sample size is needed.

We believe we have a sufficient sample size for this qualitative study. In qualitative methodology, unlike quantitative methods, there is limited value in a priori calculation of the sample size. This is because the quality of qualitative approaches does not derive from the number of participants or the extent to which they can be said to be representative of a larger population. Rather methodological rigour stems in part from purposeful sampling (in this case to elicit a range of views) and in-depth analysis to reach a compelling account of the themes which emerge from the data alongside reflexivity by the authors. We recognise that had we included patients with very different characteristics (e.g. lower health literacy) we may have found different themes, and have now more fully acknowledged and explained this in the discussion as per our response above to reviewer 1.

6. It is not clear from the manuscript whether the scenarios were discussed with participants sequentially or randomly. If sequentially, then it should be described as a potential limitation of the results. Participants' views of open-label placebo could have been negatively biased by their discussion of deceptive placebos if the open label discussion immediately followed the deceptive placebo discussion. A random presentation of the various scenarios would have been preferable.

Thank you for highlighting this limitation we have amended our text to better reflect this as follows: 'Our findings may also be limited by the sequence in which cases were presented to groups. Participant views on open-label placebo may have been influenced by preceding discussions of placebo pills prescribed deceptively. However, it was felt important to present scenarios in a way that would encourage discussion and offer participants a "way in" to this complex topic, hence we chose to present the more familiar examples of deceptive placebo prescribing before moving on to explore open label placebo.' (pg 12)

VERSION 2 – REVIEW

REVIEWER	Steven Savvas National Ageing Research Institute, Australia
REVIEW RETURNED	05-Dec-2019
GENERAL COMMENTS	thank you for your considered response to the reviewers' feedback.

REVIEWER	Patrick Finan Johns Hopkins University School of Medicine
REVIEW RETURNED	18-Dec-2019

GENERAL COMMENTS	I have read the authors' revised manuscript and response to reviews. They have done a thorough job addressing the reviews. I believe this paper will make a useful contribution to the literature.
--